# Renal Function Preservation in Purely Off-Clamp Sutureless Robotic Partial Nephrectomy: Initial Experience and Technique

**DOI:** 10.3390/diagnostics14151579

**Published:** 2024-07-23

**Authors:** Antonio Franco, Sara Riolo, Giorgia Tema, Alessio Guidotti, Aldo Brassetti, Umberto Anceschi, Alfredo Maria Bove, Simone D’Annunzio, Mariaconsiglia Ferriero, Riccardo Mastroianni, Leonardo Misuraca, Salvatore Guaglianone, Gabriele Tuderti, Costantino Leonardo, Antonio Cicione, Leslie Claire Licari, Eugenio Bologna, Rocco Simone Flammia, Antonio Nacchia, Alberto Trucchi, Riccardo Lombardo, Giorgio Franco, Andrea Tubaro, Giuseppe Simone, Cosimo De Nunzio

**Affiliations:** 1Department of Urology, Sant’Andrea Hospital, La Sapienza University, 00185 Rome, Italy; antonio.franco@uniroma1.it (A.F.); sara.riolo@uniroma1.it (S.R.); giorgia.tema@uniroma1.it (G.T.); alessio.guidotti@uniroma1.it (A.G.); antonio.cicione@uniroma1.it (A.C.); antonio.nacchia@uniroma1.it (A.N.); alberto.trucchi@uniroma1.it (A.T.); andrea.tubaro@uniroma1.it (A.T.); cosimo.denunzio@uniroma1.it (C.D.N.); 2Department of Urology, IRCCS “Regina Elena” National Cancer Institute, 00144 Rome, Italy; aldo.brassetti@ifo.gov.it (A.B.); umberto.anceschi@ifo.gov.it (U.A.); alfredo.bove@ifo.gov.it (A.M.B.); simone.dannunzio@uniroma1.it (S.D.); maria.ferriero@ifo.gov.it (M.F.); riccardo.mastroianni@ifo.gov.it (R.M.); leonardo.misuraca@ifo.gov.it (L.M.); salvatore.guaglianone@ifo.gov.it (S.G.); gabriele.tuderti@ifo.gov.it (G.T.); costantino.leonardo@ifo.gov.it (C.L.); rocco.flammia@uniroma1.it (R.S.F.); giuseppe.simone@ifo.gov.it (G.S.); 3Urology Unit, Department of Maternal-Child and Urological Sciences, Policlinico Umberto I Hospital, “Sapienza” University of Rome, 00185 Rome, Italy; leslie.licari@uniroma1.it (L.C.L.); eugenio.bologna@uniroma1.it (E.B.); giorgio.franco@uniroma1.it (G.F.)

**Keywords:** robotic surgery, enucleation, RAPN, kidney cancer, nephron sparing surgery

## Abstract

The aim of our study is to evaluate the effectiveness and safety of a sutureless off-clamp robot-assisted partial nephrectomy (sl-oc RAPN), particularly its impact on renal function. A multicenter study was conducted from April 2021 to June 2022. Patients diagnosed with a renal mass of >2 cm and a PADUA score of ≤6 consecutively underwent an sl-oc RAPN procedure. Tumor features, patients characteristics, and intraoperative outcomes were assessed. An evaluation of renal function was performed preoperatively, and again at 1 and 3 months after surgery by measuring the creatinine and blood urea nitrogen levels. The renal function of the two separate kidneys was assessed by a sequential renal scintigraphy performed before and at least 30 days after surgery. A total of 21 patients underwent an sl-oc RAPN. The median age was 64 years (IQR 52/70), the median tumor diameter was 40 mm (IQR 29/45), and the median PADUA score was 4 (3.5/5). The intraoperative outcomes included operative time (OT), 90 (IQR 74/100) min; estimated blood loss (EBL), 150 (IQR 50/300) mL; and perioperative complications, CD > 3 1(5%); only two patients presented positive surgical margins in their final histology (2/21, 10%). Compared to the preoperative value, a decrease in renal function was highlighted with a statistically significant median decrease of 10 mL/min (*p* < 0.01). The renal scintigraphy showed an overall decrease in renal function compared to the preoperative value, with a range in the operated kidney that varied from 0 to 15 mL/s and from 0% to 40%, with a median value of 4 mL/s and 12%. sl-oc RAPN is a safe procedure, with a minimal impact on kidney function alteration. This technique has proven effective in preserving renal function and maintaining optimal oncological outcomes with limited complications.

## 1. Introduction

The clinical impact of robot-assisted partial nephrectomy (RAPN) for localized renal masses is still considered controversial according to EAU guidelines. However, some studies have shown its relevant advantages over the classic open and laparoscopic techniques [1,2]. Besides the approach, two pivotal issues in the realm of RAPNs concern the management of the renal hilum and its reconstruction after tumor enucleation. The clamping of the renal artery, performed during a traditional partial nephrectomy (PN), exposes the renal parenchyma to ischemic damage and subsequent reperfusion. In fact, patients who undergo a PN may present a decrease in renal function due to both ischemia during operation and removal of the renal parenchyma.

The debate regarding whether or not to clamp the hilum persists, and despite the extensive literature available, definitive conclusions remain elusive. Antonelli et al. examined the safety profiles of different approaches by analyzing data from the first randomized trial ever conducted on the subject (CLOCK trial; NCT02287987) and concluded that no difference arose from off-clamp versus on-clamp techniques in terms of functional outcome. Furthermore, superselective clamping did not provide better renal function preservation than renal artery clamping, bringing into question the interest in this technique due to the higher risk of bleeding associated with it [3].

Classically, a single- or double-layer suture is performed after the tumor’s asportation during a partial nephrectomy [4]. Recent studies have shown that suturing the renal parenchyma may lead to possible further damage to the renal vasculature and, therefore, the loss of renal parenchyma, decreasing global renal function [5]. Therefore, to limit the ischemia time, many authors have proposed surgical techniques with zero ischemia time. While they limit the quantity of tissue removed, the use of robot-assisted techniques allows for a more precise enucleation of the neoplasm [6,7,8,9]. With the advent of robotic systems, in addition to the diversity of hemostatic materials and equipment, a sutureless off-clamp robot-assisted partial nephrectomy (sl-oc RAPN) is technically feasible, despite the little evidence in the literature that confirms its efficacy and safety [6,10,11].

The aim of our study is, therefore, to evaluate the efficacy and safety of this technique, and to evaluate its impact on renal function using a sequential renal scintigraphy.

## 2. Materials and Methods

### 2.1. Enrollment

A multicenter study was conducted from April 2021 to June 2022 at the Urology departments of the Sant’Andrea and Regina Elena IRCC hospitals in Rome. A total of 21 patients diagnosed with renal neoformation were consecutively enrolled to undergo the sl-oc RAPN technique. Exclusion criteria were patients harboring locally advanced or metastatic disease and PADUA score of >6. All patients provided their written informed consent. This study was conducted in accordance with the Declaration of Helsinki and approved by the local ethics committee: IRU study—Prot. n. 258 SA_2021. The study consisted of three distinct phases: preoperative, intraoperative, and postoperative.

### 2.2. Preoperative Assessments


–Clinical evaluation (age, body mass index, comorbidities using the Charlson score, diabetes mellitus, arterial hypertension, pathologies affecting the cardiovascular system, and ASA score);–Blood chemistry tests (creatinine and urea nitrogen) and estimated glomerular filtration rate (eGFR);–Total and separate renal function of both kidneys, assessed by sequential renal scintigraphy performed with radio—drug 99MTc—DTPA;–Imaging (abdominal CT) and nephrometric score grading (PADUA score).


### 2.3. Preoperative Preparation and Surgical Technique

A single dose of cefazolin based on the patient’s weight (2–3 g) was administered intravenously before surgery, and any anticoagulant or antiplatelet drugs were discontinued and replaced with low molecular weight heparin 7 days before. Bowel preparation was not routinely performed.

The patients were placed on the side contralateral to the lesion and transperitoneal access was carried out by positioning five robotic and videolaparoscopic ports. The optic was positioned on the pararectal line at the level of the umbilicus and two robotic ports were positioned along the midclavicular and anterior axillary lines, using the robotic instruments Hot Shears monopolar curved scissors (Intuitive Surgical, Sunnyvale, CA, USA) and ProGrasp forceps (Intuitive Surgical, Sunnyvale, CA, USA), respectively. Two laparoscopic ports (12 mm and 5 mm) for the assistant were placed in the midline, between the camera and robotic ports, creating a “U” line centered on the tumor configuration. The two accessory ports allowed for the introduction, as needed, of one or two suction irrigation devices: a Ligasure and a Weck clip applicator (Teleflex, Wayne, PA, USA). Intra-abdominal pressure of 12 mmHg was used during the entire procedure.

The Toldt’s fascia was resected and the colon was medialized. In most of the cases, direct access to the tumor was used without the need to previously identify and prepare the renal hilum. In polar tumors, the Gerota’s fascia was opened near the tumor site and the kidney had not been fully mobilized (Figure 1A). Extended kidney mobilization was performed only in posterior tumors, achieving direct visualization and better access to the neoplastic mass. The adipose tissue overlying the tumor was preserved, when feasible, to allow for correct pathological staging (Figure 1B). The margins of the tumor were marked circumferentially and incised with robotic scissors; the neoplasm was progressively separated from the healthy parenchyma following the avascular plane, and constantly checked for any bleeding. Indocyanine green may be used to evaluate the vascularization of the area surrounding the tumor in the case of complex or endophytic tumors. When bleeding vessels were observed, the forced monopolar mode was used for pinpoint coagulation (Figure 1C). Finally, the dissection of the intermediate part and the base of the tumor was always completed following the enucleation plan. Once the tumor excision was complete, repeated forced monopolar coagulation was carried out on the tumor bed until a complete dry scab was obtained (Figure 1D). To avoid eschar adhesion to the monopolar scissor, energy was administered in an almost direct contact manner, accompanied by gentle irrigation. If an incidental opening of the calyces happened, it was sutured using a 4/0 resorbable monofilament running suture. After complete tumor bed coagulation, a two-minute long surgical field inspection was routinely performed and hemostasis was further checked. A hemostatic agent (Floseal^®^) could then be applied to the tumor bed (Figure 1E). The excised tumor was removed using a 10 mm EndoCatch retrieval bag (Ethicon, Sommerville, NJ, USA). The Gerota’s fascia and the peritoneum were closed with a running barbed suture. A drain was usually left in the renal fossa for at least 24 h.

Initially, pain control was achieved using intravenous non-opioid analgesics starting from the first postoperative day (POD), with a gradual transition to oral painkillers. On POD 1, patients were fed liquid food and were gradually transitioned to a normal diet. Finally, on POD 2, the patients were mobilized and the percutaneous drainage and the bladder catheter were removed.

### 2.4. Intraoperative Assessments

The following variables were intraoperatively evaluated:
–Total operative time (OT); –Estimated blood loss (EBL); –Intraoperative complications according to Clavien-Dindo (CD) classification.

### 2.5. Postoperative Assessments 

Positive surgical margins and histology were assessed. The evaluation of renal function was performed preoperatively and at 1 and 3 months after surgery by measuring the creatinine levels, blood urea nitrogen, and eGFR. The renal function of the two separate kidneys was assessed by sequential renal scintigraphy performed before surgery and repeated at least 30 days after surgery. Subgroup analysis for tumor complexity was performed according to PADUA score and T1 classification. Postoperative complications were reported according to their CD grading.

### 2.6. Data Analysis

Data analysis was performed using SPSS.21. Data are presented in the form of mean ± SD and median (IQR). The differences between the pre- and postoperative values were assessed using the Wilcoxon test. Trifecta achievement was defined by negative surgical margins (NSM), no CD ≥ 3 grade complications, and no ≥30% postoperative eGFR reduction.

## 3. Results

A total of 21 patients who underwent an sl-oc RAPN were enrolled. The median age of patients was 64 years. The population characteristics are described in Table 1.

Neoplasms were single in 18/21 (82%) of cases. The tumors were right-sided in 10/21 (52%) cases and left-sided in 12/21 cases (57%). All tumors were categorized as T1 according to the TNM; more precisely, 10/21 (48%) were T1a tumors and 11/21 (52%) were T1b tumors. The median diameter was 40 mm (29/45) and, in one case, the patient had a single kidney. Finally, the median PADUA score was 4 (3.5/5). 

The median OT was 90 (74/100) min, the EBL was 150 (50/300) mL, and only one patient (5%) experienced a CD ≥ 3 complication. No blood transfusions were recorded among the cohort, nor were any minor complications observed. Overall, only two patients presented positive surgical margins in their final histology (2/21, 10%). With only two patients experiencing a GFR reduction of ≥ 30%, trifecta achievement was observed in 75% of the patients.

The evaluation of renal function is summarized in Table 2. The preoperative median creatinine value was 0.78 (0.70/0.81) mg/dL, the median blood urea nitrogen was 16 (14/23) mg/dL, the eGFR was 90 (60/100) mL/min, and the glomerular filtrate rate on the scintigraphy was 81 (60/100) mL/min. Compared to the preoperative value, a decrease in renal function was highlighted with a statistically significant median decrease of 10 mL/min (*p* < 0.01) (Figure 2A,B).

The renal scintigraphy results showed an overall decrease in renal function compared to the preoperative value.

The extent of the decrease in renal function in the operated kidney varied from 0 to 15 mL/s and from 0 to 40%, with a median value of 4 mL/s and 12% (Figure 2C). However, none of the patients experienced a reduction in eGFR of <30 mL/min. Further analysis was performed based on the size of the tumor. Patients with smaller tumors had a smaller decrease in renal function in the operated kidney. Indeed, patients with a T1a tumor presented a median decrease of three (1/6) vs. six (2/11) in patients with a T1b tumor (*p* < 0.05).

In the analysis of the decrease in renal function as a function of the complexity of the tumor (PADUA score), no statistically significant differences were highlighted when comparing the different PADUA classes (*p* > 0.05).

## 4. Discussion

The present study represents the preliminary experience of a promising surgical technique for the treatment of localized renal tumors. Based on our initial experience, our data suggest that a clampless and sutureless robot-assisted partial nephrectomy might represent a feasible option among surgical techniques for the treatment of localized renal tumors. Additionally, we were the first to thoroughly evaluate the containment of renal function decline in both the operated and unoperated kidneys in a sl-oc RAPN by using a renal scintigraphy, which is the gold standard for assessing renal function variation.

There is much evidence in the literature for a decrease in renal function after a PN [12,13,14,15,16] (Table 3). In patients with both kidneys, the maintenance of adequate renal function is approximately 88–91% [12]. Most studies show that the contralateral kidney only compensates for 2–6% [15]. The reason for such modest compensation by the contralateral kidney is due to the limited number of nephrons that are removed; therefore, the signals sent to the contralateral kidney are fewer in number. A similar study performed by Mir and colleagues evaluated the decline in renal function in patients undergoing a partial nephrectomy; all included patients underwent renal scanning. According to their results, 80% of kidney function was preserved [16]. In another study on 660 patients with a single kidney, the data were confirmed with a percentage of 79% [14]. In general, the overall preservation range among different studies varies from 76 to 96% [17]. In our study, the decrease in renal function in the operated kidney varied from 0 to 15 mL/s and from 0 to 40% with median values of 4 mL/s and 12%. From a pathophysiological point of view, the decrease in renal function depends either on the removal of functioning nephrons or on the lack of functional recovery after ischemia [15]. For this reason, ideally, enucleation of the renal mass should be performed as precisely as possible and adequate recovery support after the procedure’s warm ischemia should be provided. The technique adopted in our study allowed us to not clamp the renal artery and could, therefore, explain the limited loss of renal function observed in our patients. Clearly, the size of the tumor played an important role in the decrease in renal function; in fact, patients with larger tumors experienced a greater decrease in renal function (six, IQR 2/11 vs. three, IQR 1/6, *p* < 0.05).

However, how much does ischemia influence a decrease in renal function?

A study conducted on 360 single-kidney patients affected by renal neoplasia and treated with a PN correlated each minute of warm ischemia with a 6% increase in the incidence of severe de novo chronic kidney disease (CKD), suggesting a cause–effect relationship [18]. However, this and many other similar studies, did not consider all potentially relevant predictive factors such as the amount of preserved renal parenchyma. Further data demonstrated an important correlation between the ischemic interval and the amount of parenchyma saved by a PN, highlighting that ischemia time may contribute to surgical complexity. PNs burdened by greater complexity will require longer surgical times and will, therefore, lead to greater parenchymal losses. Studies that comprehensively included all relevant predictive factors, including the percentage of spared parenchymal mass, demonstrated that compared to them, ischemia time loses its statistical significance [14,20]. More recent studies have shown that functional recovery after a PN is proportional to the spared parenchymal mass and that the majority of nephrons recover almost completely from the ischemic insult after a traditional PN with the clamp technique [13]. For example, in Song’s studies, the preservation of a parenchymal mass was strongly correlated with functional recovery (*p* < 0.003), rather than with ischemia time (*p* = 0.64) [21,22].

Our study used a ‘zero-ischemia’ technique, demonstrating a limited decrease in renal function and underlined the importance of limiting renal ischemia time. In fact, some studies have evaluated the decrease in renal function in patients with zero ischemia. Ng et al. showed a preservation rate of 86% [23] and Hung et al. equally obtained a 91% rate [12]. These data are fully in line with our experience and, therefore, confirm the importance of reducing ischemia time.

It is important to emphasize that the great debate on whether to clamp or not to clamp is still alive, and although zero-ischemia procedures, off-clamp techniques, and the superselective clamping of segmental arteries directly feeding the tumor have emerged to limit ischemia, none of these techniques appear to show superior oncological and/or renal function preservation outcomes compared to the standard clamping techniques [3,19,24]. In fact, although off-clamp techniques limit or zero ischemia time, tumor bed bleeding may be profuse, causing limited visibility of the enucleation plane and thus increasing the positive surgical margin rate, the possibility for further complications, and also, in the case of massive bleeding, acute deterioration of kidney function. As a matter of fact, in the CLOCK trial, a shift to on-clamp procedures was observed in 40% of the cases [19]. However, despite the controversial data present in the literature, there is limited damage to renal function in patients treated with cold ischemia or zero ischemia, while the use of prolonged warm ischemia can lead to the development of significant renal damage [14]. Indeed, we did not experience such a complication and all of our procedures were maintained off-clamp.

As previously stated, the other major cause of renal function decline after a PN is the loss of vascularized parenchymal mass. The excision of the tumor is generally carried out including a minimum amount of healthy parenchyma with the aim of obtaining oncological radicality and, the small vessels dissected during the procedure can contribute to the devascularization of small areas of the healthy parenchyma [16]. Therefore, during a PN, a loss of function related to these processes must be considered. Several other factors may contribute to the preservation of renal function after a PN [25,26,27]. Bahler et al. suggest that the reconstruction of renal parenchyma plays a crucial role in preserving renal function after a PN [26], while Zabell et al. indicate that the volume and mass of the preserved renal parenchyma are the major factors affecting renal function [27]. In conventional PNs, two layers of sutures are typically used. The first layer involves suturing the basal area, focusing on the blood vessels and the collecting system. The second layer involves suturing the renal parenchyma. However, reducing the number of sutures, when possible, is considered a key approach to maintaining renal function. Zhao et al. demonstrated that there is no specific warm ischemia time (WIT) threshold that significantly impacts renal function; the primary determinant for renal function is the amount and quality of the preserved renal parenchyma [28]. A systematic review by Bertolo et al. found that single-layer suturing led to better outcomes compared to double-layer suturing, supporting the idea that reducing the number of sutures can preserve renal function [29]. Recently, Jin et al. compared a sutureless PN technique to the standard suturing in PNs. They found that sutureless PNs resulted in lower WIT and a reduced rate of acute kidney injury (AKI), while showing a similar decline in eGFR. Moreover, the sutureless technique demonstrated favorable outcomes in terms of OT, perioperative complications, and renal function preservation, indicating a potential advancement in the surgical treatment of small renal masses [30]. Although clear evidence is lacking, it has been shown that the use of hemostatic agents has increased in the last two decades, thus facilitating sutureless procedures. Among these, the most commonly adopted are thrombin compounds (FloSeal^®^, Tisseel^®^) and fibrin (Tachosil^®^) compounds [31,32]. In fact, although they are not always necessary, and mostly depend on the size and complexity of the renal mass, they might represent a faster and easier way to perform a reconstruction, avoiding the direct renal parenchymal damage induced by suturing. On the contrary, higher costs might be encountered when using hemostatic agents [31].

Finally, Ferriero et al. recently assessed the safety and feasibility, as well as the oncologic and functional outcomes of sl-oc RAPNs in a single high-volume center experience [10]. Their outcomes confirm our findings, demonstrating excellent bleeding control, an efficient OT, and a low rate of perioperative complications, reinforcing the benefits of sutureless PNs for treating localized renal tumors. In our cohort, the positive surgical margin rate was slightly higher (10%) than the overall rate reported in the literature (2–8%). Several factors may contribute to this, including the small sample size of our cohort. However, no significant differences have been reported in the RCTs comparing off-clamp and on-clamp procedures, demonstrating the feasibility and safety of this approach [33].

Furthermore, we tested the novel composite trifecta outcome proposed by Brassetti et al., which summarizes PN outcomes regardless of the clamping technique used, as a metric of the safety and efficacy of the procedure, including both the oncologic and functional endpoints of PNs [34].

Our study is not devoid of limitations. Certainly, the number of patients enrolled is limited, but our results represent only preliminary data which pave the way for future multicenter and randomized studies to identify the most effective surgical approach to limit renal damage. Another limitation of our research is the absence of a control group, which makes it challenging to compare our results with other techniques or to draw definitive conclusions on the topic. Finally, we have only included non-complex renal masses, due to the preliminary nature of the study. Therefore, our results must be interpreted with caution, as they might not be generalizable; however, similar studies focusing on large renal masses are ongoing. Notwithstanding these limitations, and although a long-term kidney function evaluation was missing, we were the first to test for acute renal function deterioration after a sl-oc RAPN by measuring the difference between the operated kidney and the unoperated one with a renal scintigraphy, which is the most accurate, valid, and reproducible tool used to assess kidney function after a PN.

## 5. Conclusions

An sl-oc RAPN may represent a safe and valid technique with a minimal impact on kidney function alteration. In this pilot study, the technique has proven effective in preserving renal function, maintaining optimal oncological outcomes, and limiting complications. However, the future role of this technique will have to be clarified by subsequent comparative studies and randomized control trials, which are ongoing, in order to establish which technique may represent the best surgical approach to preserve kidney function in patients affected by localized renal cancer.

## Figures and Tables

**Figure 1 diagnostics-14-01579-f001:**
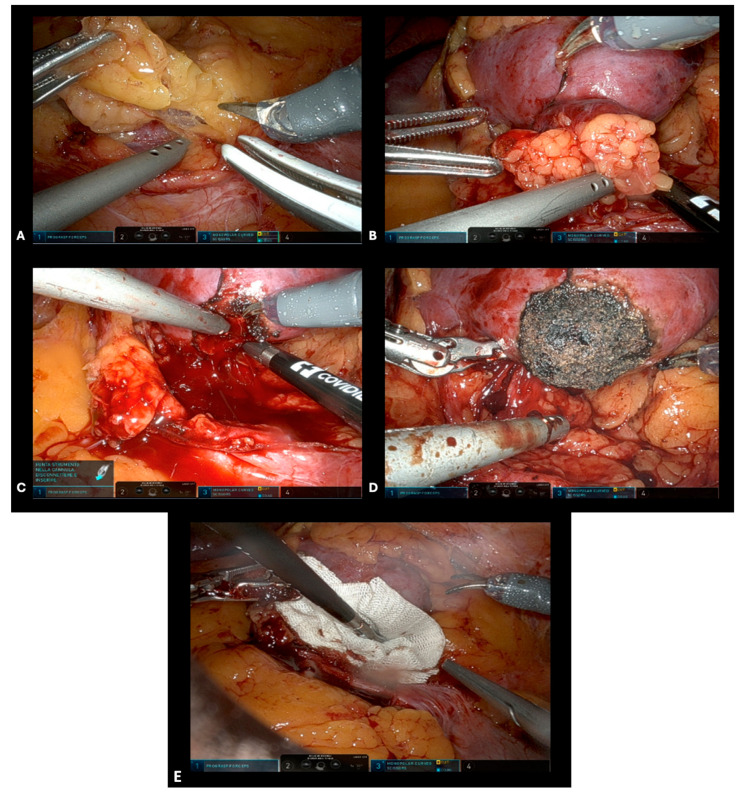
Intraoperative steps for sl-oc RAPN. (**A**) Opening of Gerota’s fascia near the tumor site. (**B**) Defining tumor margins using monopolar coagulation. (**C**) Pinpoint coagulation for bleeding vessels in the tumor bed. (**D**) Complete excision of the tumor and eschar formation by repeated forced monopolar coagulation. (**E**) Hemostatic agent application.

**Figure 2 diagnostics-14-01579-f002:**
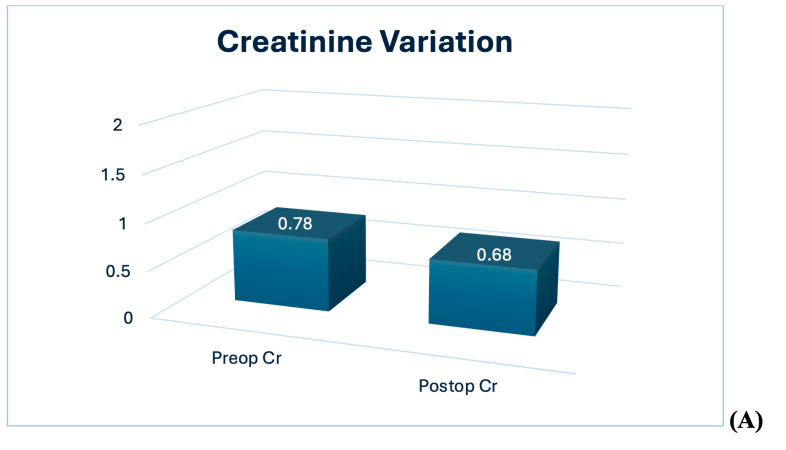
Assessment of renal function variation in both kidneys. (**A**,**B**) Creatinine and GFR decrease after surgery. (**C**) Pre- and postoperative kidney function evaluation in both kidneys by scintigraphy. Abbreviations. Cr: creatinine; GFR: glomerular filtration rate.

**Table 1 diagnostics-14-01579-t001:** Baseline characteristics and tumor features.

Variable	Median (IQR); Mean ± SD; *n* (%)
**Patients’ Characteristics**	
Age (yy)	64 (52/70) 62 ± 10
BMI (Kg/m^2^)	29 (27/31) 28 ± 3
Charlson Comorbidity Index	4 (4/5)
Arterial hypertension	11/21 (52%)
Diabetes mellitus	2/21 (10%)
Metabolic Syndrome	2/21 (10%)
Ischemic heart disease	0/21 (0%)
ASA score > 2	9/21(41%)
**Tumors’ Characteristics**	
Single	18/21 (82%)
DX SN	10/21 (48%) 11/21 (52%)
T1a T1b	10/21 (48%) 11/21 (52%)
Dimension (mm)	40 (29/45) 38 ± 10
PADUA score	4 (3.5/5) 4.1 ± 0.8

Data are reported in median/IQR and mean ± SD. Abbreviations: BMI—body mass index; ASA—American Society of Anesthesiologists.

**Table 2 diagnostics-14-01579-t002:** Renal function outcomes evaluation, *N* = 21.

	Preoperative	Postoperative 1M	Postoperative 3M	Delta (∆)
Creatinine, mg/dL	0.78 (0.70/0.81)	0.66 (0.63/0.76)	0.68 (0.66/0.74)	−0.1
BUN, mg/dL	16 (14/23)	17 (13/28)	16 (14/27)	0
eGFR, mL/min/1.73 m^2^	90 (60/100)	81 (55/90)	84 (59/90)	−10
GFR scintigraphy, mL/min	81 (60/100)	74 (56/90)	75 (58/90)	−9

Variables are expressed as a median (IQR—interquartile range). Abbreviations: BUN—blood urea nitrogen; eGFR—estimated glomerular filtration rate; GFR—glomerular filtration rate; M—month.

**Table 3 diagnostics-14-01579-t003:** Previous evidence regarding functional recovery after PN.

Author(s)	Type of Study	Sample Size	Ischemia Technique	Renal Function Outcomes
Thompson 2010 [18]	Retrospective	362	Warm ischemia	-AKI: 19% -New CKD IV: 17% -Longer WIT: OR 1.05 × min
Mir 2013 [16]	Retrospective	35 57	Hypothermia Limited warm	-GP/VS: 101% vs. 92% -%PVS: Effect 36.9 (29.5,43.3)
Greco 2019 [9]	Systematic review	22,626	Cold Warm Zero	Log^2^ GFR mean changes: −1.37 (−3.42 to 0.68) −1.00 (−2.04 to 0.03) −0.71 (−1.15 to−0.27)
Anderson 2019 [7]	RCT	40 40	Zero Warm	%GFR change: −10.7% vs. −9.4% %SRF change: −11.2% vs. −11.8%
Antonelli 2021 [19]	RCT	164 160	Zero Warm	%GFR change: −5.1% vs. −6.2% %SRF change: −2% vs. −2.5%
Present study	Retrospective	21	Zero	-%GFR change: −12% -New CDK IV: 0%

Abbreviations: RCT—randomized control trial; AKI—acute kidney injury; CKD—chronic kidney disease; WIT—warm ischemia time; GP/VS—recovery of nephron function (%GFR preservation/%volume saved); PVS—parenchyma volume saved; GFR—glomerular filtration rate; SRF—split renal function.

## Data Availability

The original contributions presented in the study are included in the article, further inquiries can be directed to the corresponding author.

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
