# Peer review of "Renal Function Preservation in Purely Off-Clamp Sutureless Robotic Partial Nephrectomy: Initial Experience and Technique"

_diagnostics, 2024, doi:10.3390/diagnostics14151579_

Round 1

Reviewer 1 Report

Comments and Suggestions for Authors

The authors interestingly addressed the effectiveness and safety of a sutureless off-clamp robot-assisted partial nephrectomy (sl-oc RAPN) and its impact on renal function. Within almost one year, they enrolled 21 patients from 2 Italian centers, harboring T1a/T1b renal tumors. Compared to the preoperative value, the authors recorded a statistically significant median decrease of 10 ml/min (p<0.01), without oncological and other functional (complications) detected. This increases importantly the amount of knowledge on this technique that could change the management of small renal masses. The overall quality of the manuscript is worthy of publication, sounding interesting and novel. The tables and Figures are clear and the manuscript reads very well. 

Several minor points warrant a comment.

-Did any patients have a history of heart valve replacement (PMID= 38526833)? Did any patients exhibit a history of chronic kidney disease for who the surgical intervention had shifted their disease to a higher Stage (for instance from stage 1 to stage 3a)? 

- More details on the role of hemostatic agents should be added to better contextualize the feasibility of the sutureless technique compared to the standard renography (PMID= 25139104, 38157157) 

Author Response

Please see attachment below

Reviewer 2 Report

Comments and Suggestions for Authors

48: Authors should specify what the two techniques are.  

67-68: A cohort of 21 patients is a small sample size. Although the authors do mention this limitation in the discussion section, the sample size has large implications on generalizability and statistical power of the findings. 

136-140: What was the rationale for postoperative performance of scintigraphy 30 days after surgery, while only measuring creatinine and BUN only until 3 months after?

How did the authors select which patients to include in the study? Was it consecutive patients, or those with more favorable PADUA, or perhaps just those with tumors >2cm. There may be selection bias in favor of "easier" tumors since it was not randomized. It is a strength that the authors provided the PADUA score, but the average score seems to reflect low complexity, which hampers generalizability of the findings.

Results: 

Figure 2: 

Figure 2B: “Preop GFR operato” should be corrected to “preop GFR operated kidney”

Figure 2C: “Preop GFR” should be corrected to “preop GFR operated”

167: There doesn’t seem to be any description of the preoperative and postoperative GFR evaluation in the “not operated” arm. How were confounding variables that could affect GFR controlled for in the “not operated” group? 

Presenting this data graphically seems to provide little contribution, especially in ascertaining statistical significance. It would likely be better presented in table format or simply in the text.

171-172: Including data and images of renal scintigraphy preop and postop would be helpful, if not necessary, to assess overall decrease in renal function and loss of renal parenchyma. The author’s themselves note that “renal scintigraphy is the most accurate, valid and reproducible tool to assess kidney function after PN.” 

Overall, it is glaring that the study does not compare renal function in sl-oc RAPN to a control (traditional RAPN that is not suture-less and clamp-less). Although the authors note this limitation in the discussion section, they also claim that their data suggests “that a clamp-less and suture-less robotic-assisted partial nephrectomy is a safe and effective treatment for localized renal tumors.” Conclusions about the safety and efficacy of sl-oc RAPN cannot be made without a comparison to traditional RAPN. 

Discussion: 

183-185: The discussion indicates that this procedure is safe and effective. As mentioned before, conclusions about the safety and efficacy of sl-oc RAPN cannot be made without a comparison to traditional RAPN. Additionally, what are the standard perioperative and postoperative metrics that indicate a “safe and effective treatment for localized renal tumors”? What is the impact of the sl-oc-RAPN modality on the applicability of these metrics (which are likely based on and pertain to traditional RAPN).

 The positive margin rate of 10% in this study is somewhat higher than the average of ~7% reported in the literature, which needs to be acknowledged by the authors.

It is intriguing for an off clamp and no suture technique. But authors should also report postop complications like AV fistula, pseudo aneurysm , bleeding, and collecting system injury. They only focus on post-op creatinine level.

Do the comparisons between this study and others control for differences in patient disease burden? Additionally, it is especially challenging to continue making comparisons between the results of this study and other studies that have larger cohort sizes and potentially different patient baselines and intraoperative complications. 

Comments on the Quality of English Language

Author Response

Please see attachment below.

Reviewer 3 Report

Comments and Suggestions for Authors

Following are suggestions for the authors to improvise the manuscript:

1. The Ethical approval numbers should be mentioned in the methods section 

2. The authors should include statistical tools and perform statistical analysis on the data.

3. The bar charts should be plotted with tools like Origin for better representation. 

4. A Table with comparison of outcomes of the previous studies from literature and the present study should be included in the Discussion section 

5. Conclusion is very generic the authors should be more comprehensive and elaborate in more detail about the conclusions derived from the study.

6. The introduction section should be improved with reference to the previous studies. Atleast 10-15 studies from the literature should be cited which gives direction to the present study. 

Overall authors need to include the above suggestions and improvise manuscript for further consideration. 

Author Response

Please see attachment below
